# Landscape Agroecology: Methodologies and Applications for the Design of Sustainable Agroecosystems

**Miguel A. Altieri** [1,*]**, Clara I. Nicholls** [2]**, Manuel González de Molina** [3] **and Angel Salazar Rojas** [4,5,*]

1. Department of Environmental Science, Policy and Management (ESPM), University of California, Berkeley, CA 94720, USA
2. International and Area Studies, University of California, Berkeley, CA 94720, USA; nicholls@berkeley.edu
3. Laboratory of the History of Agroecosystems, Universidad Pablo de Olavide (UPO 4), 41013 Seville, Spain; mgonnav@upo.es
4. Universidad Católica del Maule, Talca 3460000, Chile
5. Programa Doctorado Medio Ambiente y Sociedad UPO 4, Universidad Pablo de Olavide (UPO 4), 41013 Seville, Spain
* Correspondence: agroeco3@berkeley.edu (M.A.A.); adsalazar@ucm.cl (A.S.R.)

**Abstract:** Agroecosystem function is related to the positioning of the agroecosystem and its connectivity relationship with the surrounding landscape. Herein, three methodologies are presented, which allow assessment of the links between agroecosystems and the surrounding matrix, yielding information for promoting patterns and mechanisms that foster biodiversity and the provision of multiple ecosystem services such as biological pest control, as well as energy flows and material exchanges. The three methodologies are complementary when assessing agrolandscape-level interactions in situations of regional agroecological transition. Through the use of 11 indicators, a methodology (Assessment of Beneficial Insect Habitat Suitability-ABIHS) was applied in two northern California vineyards to determine whether each agrolandscape provided suitable environmental opportunities to sponsor biological insect pest control. The Main Agroecological Structure [MAS] applied in Chilean family farms elucidates some of the relationships between farms and their biophysical environment, generating data to analyze the links between agroecosystem landscapes, management practices, and insect diversity in family farms. Social Agrarian metabolism (SAM) applied in Spanish agrolandscapes quantifies the biophysical and energy flows in agricultural systems, testing whether such flows are capable of reproducing and/or improving fund elements such as soil, biodiversity, and landscape vegetation in successive production cycles. The three methodologies provide key information for the design of sustainable agroecosystems in the context of an agroecological transition.

**Keywords:** agroecology; biodiversity; biological control; agrarian metabolism; sustainable agriculture





## 1. Introduction

Evidence from around the world indicates that agriculture's negative impacts on biodiversity increase as production intensifies through the expansion of large scale monocultures, removal of non-crop habitat and increased input of pesticides and fertilizers [1]. The simplification of landscape natural habitats and the decline of biodiversity affects the functioning of agroecosystems by reducing the provision of multiple ecosystem services [2]. In order to restore such services and optimize agroecosystem function, the area of suitable natural or semi-natural habitat should be maximized in agrolandscapes, balancing crop yields and biodiversity conservation. Some researchers have found that species richness falls precipitously once habitat area falls below about 30% [3], suggesting that at least 20% of the native habitat needs to be retained in most agrolandscapes in order to support the provision of many of nature's contributions to crop productivity and stability [4].

Many agroecologists therefore suggest that a moderately heterogeneous matrix of natural vegetation in agrolandscapes is essential for promoting mechanisms that foster

biodiversity and the provision of ecosystem services such as biological pest control, soil biological fertility, and resilience in agricultural fields [5]. As in the case of many small farms, the ideal is to have a high-quality landscape matrix within which fragments of high-diversity native vegetation persist along with biodiverse agroecosystems [6]. A recent meta-analysis covering 122 studies conducted in Asia, Europe, and North and South America reveals consistently positive effects of crop and landscape heterogeneity on plant, invertebrate, vertebrate, pollinator, and predator biodiversity [7]. Such agrolandscapes are still common in many regions dominated by small-scale agriculture characterized by polycultures and agroforestry systems surrounded by hedgerows, grassy borders, and forest patches, generating positive co-benefits for production, biodiversity, and local people [8].

Given the socio-ecological impacts of industrial agriculture and its vulnerability to climate change, a growing number of scholars and policymakers are calling for a transition from input-intensive monocultures to more biodiverse, low-input farming systems [9]. One requirement of such conversion is to increase plant diversity in agroecosystems in the form of rotations, intercropping, enriched field borders, etc, which promote ecological interactions which in turn favor crop yield stability, pollination, weed suppression, and pest suppression [10]. In addition, agroecosystems consisting of mosaics of crops, livestock, and forests can foster critical functions such as maintaining water quality, regulating water flow, recharging underground aquifers, mitigating flood risks, and moderating sediment flow [11]. All these benefits are expected to be greatest in agroecosystems surrounded by a complex landscape matrix with strong connectivity relationships with semi-natural habitats [12].

Since complex socio-ecological nature-agriculture interdependencies emerge in biodiverse agrolandscapes, there is a need to analyze and assess the ecological relationships between agricultural systems and the surrounding matrix, including the exchange of energy and materials between a given agroecosystem and its environment. Such information is key for the design of sustainable agroecosystems that are resilient and dependent on ecological processes rather than external inputs in the context of an agroecological transition at the territorial level.

A line of inquiry has been to elucidate landscape factors that mediate arthropod abundance and species richness differences between farms surrounded by simple or complex habitats, and how landscape biodiversity enhancement can improve the functioning of natural pest control [13]. The first methodology presented herein (Assessment of Beneficial Insect Habitat Suitability-ABIHS) assesses whether a particular landscape matrix is conducive to enhancing natural enemies of insect pests, utilizing simple indicators that, depending on their score, can suggest habitat improvements in the form of specific crop/vegetation designs and management practices.

To visualize agroecological relationships established between farms and their biophysical environment, a second methodology called Main Agroecological Structure (MAS) has been proposed as an environmental index using metrics of composition, configuration, connectivity, and heterogeneity of landscapes surrounding agroecosystems [14]. Since the transition to more sustainable agri-food systems implies laying out new agroecological territories where an increased number of diversified farms are integrated with the landscape, it is important to assess the closure of basic biophysical cycles at the territorial level [15]. The third methodology, Social Agrarian metabolism (SAM), is a tool to assess agricultural sustainability by quantifying the biophysical flows, i.e., energy, macronutrients, and carbon of agricultural systems, and by testing whether these flows are capable of reproducing and even improving fund elements such as soil, biodiversity, and woodland in successive production cycles [16].

### 1.1. Landscape Diversity and Biological Pest Control

It is well documented that diversification of cropping systems at the field level can lead to enhanced biological regulation of insect pests, as in diversified farms natural enemies

tend to be more abundant and efficient than in monocultures [17,18]. Effects of plant diversification on insect populations extend beyond the field level, as research shows how landscape spatial heterogeneity through increased diversity of crop and non-crop cover types in agricultural landscapes benefits biological control of pests [19]. The general consensus is that crop fields surrounded by complex landscapes versus simple landscapes exhibit higher natural enemy abundance and species richness as well as lower pest pressure. A review documented the effects of landscape composition on natural enemies in 24 studies showing that landscape complexity enhanced natural enemy populations in 74% of the cases [20]. Despite such robust evidence, researchers debate to what extent landscape composition vs. configuration affects pest-natural enemy interactions at different scales. Evidence indicates that small fields with irregular shapes bordered by seminatural margins exhibit greater biocontrol services than large fields with few or no margins [21].

Available information from landscape studies has been valuable in guiding agroecosystem designs aimed at enhancing pest regulation mechanisms. For example, in vineyards where natural enemy abundance is usually higher in fields close to weedy hedgerows or forests [22], the creation of corridors that connect vines with a semi-natural habitat is an interesting strategy. In northern California a corridor composed of a mixture of succession flowering species connected to a riparian forest that cut across an organic monoculture vineyard, enhanced timely circulation and dispersal movement of predators from the forest into the center of the field, leading to reduced pest densities up to 50 vine rows away from the corridor [23].

Landscape heterogeneity is also related to the diversity of crop species deployed in a geographical area. Creating a crop mosaic by increasing the number of crop types in an agrolandscape may enhance natural enemies sustaining biological control processes [24]. This depends on the spatial scale, particularly on the life histories and movement capabilities of different natural enemies. For example, specialist parasitoids often perform better in smaller spatial scales than larger predators [25]. Crop diversity enhanced aphid regulation in fields with mosaics of various crops compared to low-crop-diversity landscapes [26]. These studies suggest that many pests can be reduced by optimizing the composition, configuration, and temporal heterogeneity of the crop mosaic in a landscape. However desirable effects may be hard to replicate year after year if the spatial and temporal heterogeneity of crops changes in a given landscape.

### 1.2. A Simple Methodology to Assess Beneficial Insect Habitat Suitability at the Agrolandscape Level (ABIHS)

Most studies documenting the effects of landscape diversity on insect pests and natural enemies have been conducted by selecting independent gradients of landscape-wide crop diversity and fields exhibiting various configurations of semi-natural vegetation borders with variable floral composition. Research plots usually consist of fields of contrasting landscape matrices (simple versus complex) where crop diversity, semi-natural habitat cover, and natural enemy/pest densities are estimated using various sampling methods [27]. Research results document the effects of landscape composition on interactions between natural enemies and insect pests, population densities of predators and herbivores, parasitization rates, and seldomly levels of crop damage. Although useful, results do not necessarily lead to defined crop and landscape diversification schemes that farmers can implement. Therefore, there is a need to provide practical tools that farmers can use to assess whether the prevalent crop/habitat composition and configuration of the agrolandscape is conducive to enhancing biocontrol under their agroclimatic conditions, and if not, what agroecological designs may be needed to improve habitat quality for beneficial insects.

Assessing beneficial insects' habitat at the field and landscape levels requires evaluation of landscape (i.e., presence of woodlands, riparian forests, etc) and farm (crop diversity, rotations, cover crops, etc) features, and the dominant agricultural practices utilized (use of pesticides, conventional versus organic practices, etc), which determine the conditions for natural enemies to thrive or not. Based on a practical field guide [28], the herein proposed

methodology (Assessment of Beneficial Insect Habitat Suitability-ABIHS) provides a set of indicators that farmers can apply through a series of field observations and simple measurements (i.e., the number of crop species and varieties grown in time and space, soil and pest management practices, as well as observations of the landscape matrix surrounding the farms).

The indicators described in Table 1 can be assessed and ranked separately and assigned a value between 1 and 5 according to the criteria described in Table 1 (1 being a poor value for habitat suitability, 2.5 a moderate value, and 5 indicating a high or good value). Once the indicators are applied, each farmer can visualize the overall quality or suitability of their habitats for enhanced biological control in an amoeba diagram. By applying the same methodology simultaneously to several farms in a particular region, it is possible to visualize which farms are closer (or farther) to what can be considered an agroecological optimum. Scores obtained from the rapid observations permit farmers to determine which indicators are performing poorly and make design and management decisions to improve such indicators, thus enhancing overall agrolandscape habitat quality. The methodology requires farmers' participation in the selection and validation of indicators, particularly in defining common criteria on how to rank each indicator. As measurements are based on the same indicators, the methodology can allow quick comparisons to reveal differences between various farms but also allow farmers to monitor the evolution of their agrolandscape habitat quality along a timeline. The method is flexible and applicable to a wide assortment of agroecosystems in a series of geographical and socio-economic contexts.

**Table 1.** Indicators and evaluation criteria to assess the habitat suitability of a particular agro-landscape for natural enemies that regulate insect pest populations (Altieri and Nicholls, unpublished data).

| Indicator | Value | Evaluation Criteria |
|---|---|---|
| **Landscape Level** | | |
| % of farm area with natural (N) or Semi-natural area (SN) | 1<br>2.5<br>5 | <10%<br>10–30%<br>>30% |
| % of fam perimeter with N or SN habitat | 1<br>2.5<br>5 | 0–10%<br>10–30%<br>>30% |
| Plant diversity composition of N and SN habitats | 1<br>2.5<br>5 | Hedgerow or weed patches composed of one or two species (naturalized or invasive)<br>Mix of 3 to 5 native or naturalized species<br>>5 native and naturalized plant species |
| % of vegetation composed of wildflowers and flowering shrubs and trees | 1<br>2.5<br>5 | <20%<br>20–50%<br>>50% |
| % of plant species flowering in early, mid, and late crop season | 1<br>2.5<br>5 | <20%<br>20–50%<br>>50% |
| **Farm Level** | | |
| # crops species deployed in the farm area in various fields or plots (crop mosaics) | 1<br>2.5<br>5 | One or two crops<br>2–5<br>>5 |
| Crop spatial diversity | 1<br>2.5<br>5 | Monoculture<br>2–3 species intercropped<br>>4 species intercropped |
| Crop temporal diversity | 1<br>2.5<br>5 | No rotation, no vegetative fallow<br>One crop rotation per year, with or without fallow<br>>2 rotations per year, including legume crop, with fallow |

**Table 1.** *Cont.*

| Indicator | Value | Evaluation Criteria |
|---|---|---|
| **Management Level** | | |
| Pesticide use | 1 | Frequent use of chemical insecticides and herbicides |
| | 2.5 | Use of microbial or botanical pesticides |
| | 5 | Reliance on practices that encourage biological control |
| Provision of flowering resources | 1 | No flower provisioning |
| | 2.5 | Provision of 1–2 flower species dispersed in the field |
| | 5 | Provision of 3 or > flower species along borders or strips within fields |
| Practices to provide shelter | 1 | No practices |
| | 2.5 | Use 1–2 practices (i.e rock piles, dispersed shrubs) |
| | 5 | >3 practices (i.e., fallows, rock piles, undisturbed ground, mulch, dead wood piles, etc.) |

Figure 1 shows a comparison between two vineyard farms located nearby in northern California (Altieri and Nicholls, unpublished data), A: diversified organic vineyard with a moderately diverse landscape matrix and B: monoculture vineyard in transition to becoming organic surrounded by a vegetation-poor landscape. Results from the rapid assessment presented in the amoeba diagram show values of 11 indicators applied in the two vineyards. Farm A exhibited values above 2.5 for most indicators, as it featured landscape and crop level diversification schemes; however, some indicator values, such as crop rotation, could be improved by adopting summer cover crops in addition to the winter cover crops used. The indicator of the presence of flowers year-round in the surrounding habitat could be improved by enriching the floral composition of field borders. As the indicators suggest (all values under 2.5), the habitat of farm B is far from being considered suitable, with significant room for improvement regarding redesign and agroecological management practices conducive to enhanced biological control, as it lacks landscape and crop plant diversity and soil cover. Major interventions are required to improve landscape and farm agroecological features, including the elimination of organic pesticides, use of cover crops, planting of flowering hedgerows, and extending them to surround the whole farm perimeter. Such farm design should increase positive interactions between soil, plants, and insects, thus promoting biological pest control while minimizing the use of pesticide inputs.

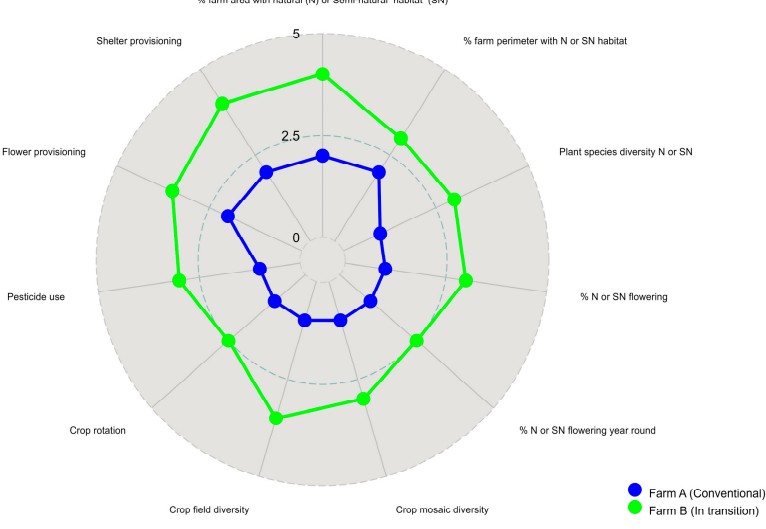

**Figure 1.** Comparison of agrolandscape habitat suitability indicators for natural enemies in two contrasting California agroecosystems [Altieri and Nicholls, unpublished data].

## 2. The Main Agroecological Structure (MAS): Assessing Interactions Between the Matrix of Semi-Natural Vegetation with Agroecosystems

MAS is a methodology that helps in the analysis of the spatial and functional organization of the matrix of semi-natural elements in interaction with the agricultural structure [29]. MAS provides information essential to promote agroecological schemes yielding critical insights that must be taken into account when designing agrolandscapes to foster biodiversity and the provision of multiple ecosystem services [30–32]. By employing metrics pertaining to the composition, configuration, and heterogeneity of the landscapes that encircle agroecosystems, MAS provides an environmental index that encompasses both ecosystemic and cultural parameters, thereby facilitating the visualization of key interactions between farms and their biophysical surroundings [14].

MAS was utilized to analyze the agrarian landscape of the Chilean Mediterranean, home to approximately two million rural inhabitants occupying 80% of the area [33,34]. Recent geopolitical transformations have significantly altered the region's landscape matrices, agrarian structures, and social dynamics, leading to economic disruption and increased social–environmental risks [35–38]. Evidence indicates that agricultural sustainability is linked to the management of both cultivated and uncultivated landscape diversity. Agroecological transition in larger farms surrounded by a simple matrix tends to be slower and more troublesome than in smallholder agroecosystems employing agroecological practices within a moderately heterogeneous matrix [4,39]. MAS-generated data to analyze the links between agroecosystem landscapes, management practices, and insect diversity in family farms near Cauquenes, Chile herein presented (Salazar Rojas, unpublished data).

### 2.1. Applying MAS to an Agrarian Landscape in the Chilean Mediterranean

The agricultural surface of the Maule region of Chile is dominated by agroexport crops that are highly industrialized, including (65%) forest pine/eucalytus plantations, (14%) fruit plantations (cherries, European hazelnuts, apples, and walnuts), and (8%) vineyards, occupying 87% of the total arable land, while the rest of the area is composed of small farms with small orchards, leguminous crops, vegetables, tubers, and home gardens [40]. To quantitatively and qualitatively measure agrobiodiversity, particularly in terms of structure, the following indicators were utilized (Table 2).

**Table 2.** MAS indicators: metrics evaluated, description, and method (based on [29]).

| Parameter | Description | Method |
|---|---|---|
| Connection with the main ecological landscape structure [CMELS] | Assesses the distance of the farm in relation to the nearby fragments of natural vegetation, mainly forest covers and bodies of water. | GIS/focus group |
| Extension of external connectors [EEC] | Evaluates the percentage of the linear extension of live fences located in the perimeter of the farms. | GIS/focus group |
| Extension of internal connectors [EIC] | Internally evaluates the percentage of the linear extension of the rows of vegetation. | GIS/focus group |
| Diversification of external connectors [DEC] | Evaluates the diversity of live fences or hedges located in the perimeter of the major agroecosystem. | GIS/focus group |
| Diversification of internal connectors [DIC] | Evaluates the diversification of internal live fences. | GIS/focus group |
| Soil Use and Conservation [USC] | This parameter evaluates the distribution percentage of different covers within the farm and the conservation of the soil (evidence of erosion). | GIS/Interview/focus group |
| Management of Weeds [MW] | Evaluates the management practices and systems of weeds. | Interview/focus group |

**Table 2.** *Cont.*

| Parameter | Description | Method |
|---|---|---|
| Other Management Practices [OP] | Is an indicator that expresses the type of production system (ecological, conventional, or in transition) of each farm | Interview/focus group |
| Perception-Awareness [PA] | Evaluates the degree of conceptual clarity and awareness of producers regarding agrobiodiversity. | Interview/focus group |
| Level of Capacity for Action [CA] | Evaluates the capacities and possibilities of farmers to establish, maintain, or improve their MAS | Interview/focus group |

In order to understand the connection between the types of agroecosystems in the Maule region and the surrounding landscape, MAS uses the following Equation (1).

$$MAS = CMELS + EEC + EIC + DEC + DIC + LU + WM + OP + PC + CA \qquad (1)$$

Surveys were conducted in each of the agroecosystems studied (N = 67), which included field observations and questionnaires applied to farmers [41]. Data obtained was analyzed using a generalized linear model (GLM), where a Poisson distribution for the count variables of predator and parasite abundance is assumed. All graphs and analyses were carried out using R version 4.3.2. and its packages "ggplot2" and "stats" [42,43].

*2.2. Landscape Structure, the Presence of Native Vegetation Patches, and the Response of Natural Enemies*

As described in Table 3, the surface of the study area was 48.98 $\pm$ 79 ha (mean $\pm$ sd), which showed a moderate presence of semi-natural vegetation patches, representing 5.5 $\pm$ 12.1% (mean $\pm$ sd) of the monitored surface, constituting important semi-natural areas for the maintenance of auxiliary entomological fauna at the landscape level [44]. The indicators obtained for the degree of connectivity with the agroecosystems studied are represented by the distances calculated between vegetation patches of 92.59 $\pm$ 73.88 m (mean $\pm$ sd), the distances of these vegetation patches to the center of the farm of 102.8 $\pm$ 55.1 m (mean $\pm$ sd), suggesting high connectivity according to Leon-sicard et al. [45]. This is also influenced by the large extension of these vegetational patches of 507.5 $\pm$ 242 m (mean $\pm$ sd), with a diversity that varied between 5–10 spp. distributed in 2–3 plant layers. This highlights the importance of plant connectors at the periphery of agroecosystems as they facilitate the movement of functional agrobiodiversity into productive systems [46]. The average condition of the agroecosystems is 64.6 $\pm$ 9.4 (see Figure 2a), indicating that the connection between agrobiodiversity and surrounding native vegetation is slightly to moderately developed [14,47].

**Table 3.** Parameters evaluated in the agroecosystems (n = 12) [Salazar Rojas, unpublished data].

| | Agroecosystems Assessment |
|---|---|
| Area cultivated (ha) | 1.17 $\pm$ 1.1 |
| Parameters | |
|     Area of influence (ha) | 48.98 $\pm$ 79 |
|     Semi-natural habitat patches (%) | 5.5 $\pm$ 12.1 |
| CMELS | |
| • *Distance between SNH patches (m)* | 92.59 $\pm$ 73.88 |
| • *Distance between SNH patches and center of agroecosystem (m)* | 102.8 $\pm$ 55.1 |

The insect communities identified as taxonomically recognizable units were dominated by species of the orders Hemiptera (55.4%), Diptera (35.2%), Coleoptera (5.5%), and Heminoptera (2.8%), mostly predators and parasitoids of the main pests that attack the

crops of the area, including aphids, leafhoppers, and moths. Figure 3 shows the results of the models simplified, where it can be observed that the abundance of predators was positively influenced by the MAS index ($x^2$ = 51.25; df = 1; p = 0.008) and parasitoid abundance was slightly, but not statistically, influenced by MAS ($x^2$ = 0.24; df = 1; p = 0.62). These results show how biodiversity patterns are influenced by MAS, particularly the influence of the complexity of the surrounding landscape on natural enemies, which can provision natural pest control services under scenarios of a well-structured matrix [48].

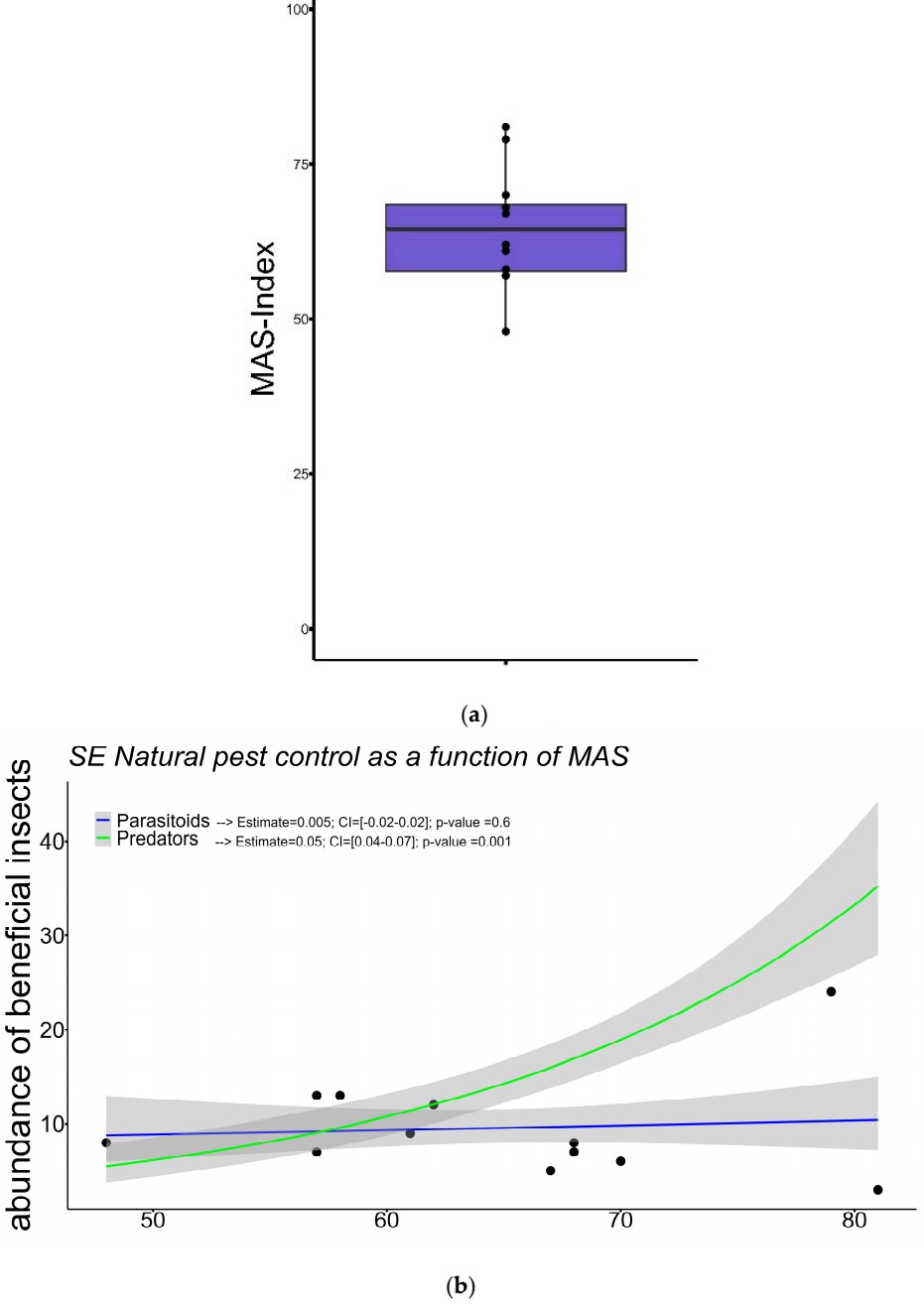

(**a**)

(**b**)

**Figure 2.** (**a**) Assessment MAS-Index boxplot. (**b**) Relationship between Main Agroecological Structure of all agroecosystems and Abundance Predators (green), Abundance Parasitoids (blue) [Salazar Rojas, unpublished data].

By allowing researchers and farmers to observe the relationship between landscape structure, the presence of native vegetation patches, and the response this generates on

natural enemies, MAS aids in understanding the positive effects of maintaining and increasing areas of natural and semi-natural vegetation within the agroecosystem and its surrounding perimeter.

### 3. A Metabolic Approach to Agricultural Landscapes: Assessing Energy, Material, and Information Exchange Between Agriculture and Its Socio-Ecological Environment

Social Agrarian Metabolism (SAM) adapts Social Metabolism methodology to agriculture [49,50], utilizing an agroecological perspective [51]. The hybridization between Agroecology and Social Metabolism enables the design and fine-tuning of a theoretical and methodological tool capable of analyzing any agrarian system from the integrated and multidimensional perspective of sustainability.

SAM assesses the exchange of energy, materials, and information between the agricultural sector and its socio-ecological environment (see Figure 3). The main aim of the metabolic process is the growing and appropriation of plant biomass (net primary productivity) from the land in order to directly or indirectly satisfy crops, livestock, and raw consumption human needs. Every agroecosystem is endowed with fund elements: biophysical funds such as levels of soil fertility, biodiversity, water, organic matter, etc., and social fund elements such as means of production, human labor, etc.; both are interlinked and make the production of biomass and the provision of agroecosystem services possible. Such funds are fed by energy and material flows that are consumed or dissipated during the metabolic process, such as fertilizers, seeds, energy, etc. The fund elements use these inputs to transform them into goods, services, and waste. The economy's ultimate goal is not the production and consumption of goods and services, but the reproduction and improvement of the processes necessary for their production and consumption [52]. Therefore, attention should shift away from energy and material flows and instead focus on whether fund elements are improved or at least reproduced during each productive cycle. In other words, the focus switches from the production and consumption of goods and services—what conventional economics and agronomy focus on—to sustainability, and whether both production and consumption can be maintained indefinitely.

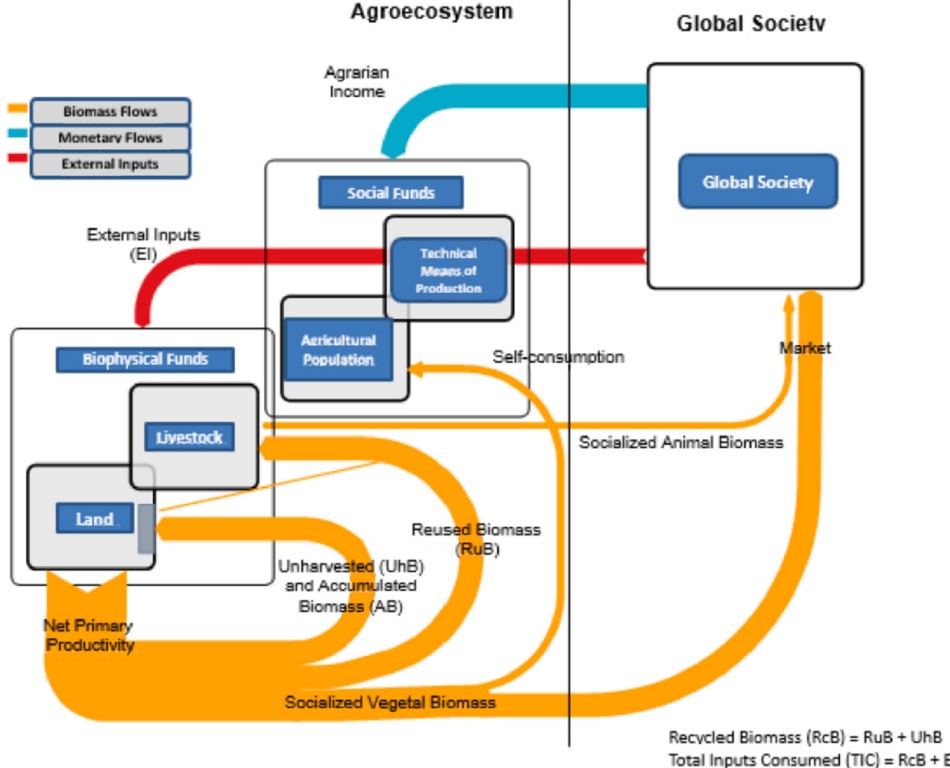

**Figure 3.** Flowchart of Social Agrarian Metabolism. Source: [51].

SAM analyzes the role of energy flows within agroecosystems as a significant part of the biomass generated that must recirculate in order to perform the basic productive and reproductive functions of the agroecosystem. Therefore, the sustainability of agroecosystems correlates positively with the quantity and quality of its internal loops and exchanges with the landscape matrix and the energy flows that circulate within it, and from the matrix whose function is to reproduce the fund elements [53,54]. So when the surrounding landscape and internal biodiversity of a monocultural agroecosystem are reduced, external and internal loops diminish, needing to generate internal order through the import of significant amounts of energy via external inputs, seriously compromising its level of sustainability. The maintenance of internal loops in agroecosystems is directly related to the use of a significant part of net primary production to fuel them. This has major implications when it comes to calculating Net Primary Productivity (NPP), which must then be broken down into different categories according to its productive or reproductive functionality. A new methodology [55,56] allows researchers to know whether the flows that enter and recirculate within agroecosystems are capable of reproducing the biophysical fund elements, or whether, on the contrary, these elements are being compromised.

SAM also considers information flows within the Social Metabolism, allowing the integration of social and economic dimensions of sustainability [57,58]. The monetary flows received by farmers for selling their products usually inform decision-making, therefore determining the reproduction—or not—of social fund elements (human labor and technical capital). Below, two examples of the application of this socio-metabolic approach to the case of Spanish agriculture in the 20th century are presented; results come from previous research cited below.

### 3.1. The Landscape as a Socio-Metabolic Footprint

Each specific arrangement of the agroecosystem is reflected in a specific organization of the landscape which imposes a particular footprint on the territory [59–61]. For example, in organic metabolic regimes [62], agroecosystems function in an integrated manner in such a way that the internal loops clearly extend beyond the cultivated land and cover surrounding environments. In the past, most of the energy and materials in agroecosystems came from domestic extraction, and very little was imported. Therefore, agroecosystems had to maintain a strict balance between the different uses of the territory; for example, the conservation of forest patches to provide timber and firewood for building and heating houses, the restriction of arable land to maintain grazing land to feed livestock, etc. The increase in entropy that came about with agricultural intensification was usually compensated by the import of nutrients, generally through livestock (manure), and other low-entropy areas in the agrolandscape such as pastureland or woodland. The landscape heterogeneity and agrosilvopastoral integration were key to the structuring of the different loops that captured, stored, and transferred energy.

On the contrary, the landscapes of industrialized agriculture are simplified to the same extent that the internal loops within their agroecosystems are reduced [63]. The Spanish case is a paradigmatic example. The changes caused by the industrialization of agriculture profoundly affected landscapes via the breakdown of synergies between different land uses, the dramatic decrease of hedgerows, the simplification of crop rotations, the expansion of monoculture, the separation of livestock from cropland and pastureland, the expansion of forest plantations at the expense of natural ones excluding other non-forest uses, etc. The breakdown of internal circuits increasingly forced the use of external inputs, changing the metabolic relationship with the territory. Agricultural production grew essentially in two main ways: the total amount of biomass per hectare and the percentage of the net primary productivity appropriated by society [64,65]. Among all extracted biomass components, biomass from primary crops (grain from cereals, fruit from fruit trees, etc.) almost tripled. In turn, the bulk of extracted biomass is concentrated on specialized crops of greater commercial value (olive groves, fruits, and vegetables). Since the 1960s, and more intensely since the 1990s, the Spanish agricultural sector has intensified

stockfeed production to partially sustain livestock specialization. Both production and technological efforts have been directed towards maximizing the share of biomass of the highest commercial value, entailing the reduction of crop multifunctionality. In other words, agricultural production growth has been much greater than the growth of agroecosystems' net primary productivity [51].

Spanish agriculture's industrialization led to increasingly segregated land uses and the loss of functional synergies typical of agrolandscapes with agrosilvopastoral integration. Livestock grew progressively on landless farms (as they depended on forage and feed brought from imports), breaking the close ties between agricultural and livestock activities, making the replenishment of soil fertility with animal manure and the use of animal traction difficult. Therefore, the introduction of synthetic chemical fertilizers and mechanization were necessary, increasing the use of external inputs coming from fossil fuels and depressing the energy efficiency of agricultural activity. The same phenomenon affected forest lands dedicated to forestry plantations or conservation areas, restricting traditional uses of the landscape such as grazing, gathering, firewood collection, etc.

Such agrolandscape simplification has compromised the reproduction of agroecosystem fund elements, particularly biodiversity levels, which is problematic given its strong relationship with the productivity of agroecosystems (measured in terms of total biomass) [53]. Biodiversity expresses the link (complex food chains) between low entropy and dissipative structures: some types of biomass feed others and vice versa, and ecosystems with larger amounts of energy entering the food web will be able to support longer food chains and hence greater biodiversity [66]. In the particular case of agroecosystems, different authors have found that the incorporation of forage crops in rotations is one of the drivers of the biodiversity increase associated with the conversion of conventional farms into organic farms [67–69]. The measurement of energy efficiency through indicators such as Energy Return on Investments (EROIs) can reflect the interconnection of agroecosystem internal cycles. Biodiversity EROI [70] provides useful information on the extent to which energy invested in the agroecosystem contributes to sustaining the food chains of heterotrophic species. "Biodiversity EROI" has been estimated for Spanish agriculture over the last hundred years (2).

$$Biodiversity\ EROI = UhB/TIC \tag{2}$$

where UhB is the non-harvested portion of the net primary productivity and TIC is total input consumed to produce it.

Biodiversity EROI decreased by 14%, from 0.86 in 1900 to 0.74 in 2010, indicating a decrease in UhB in relation to TIC, which entails a lower level of relative energy availability for wild heterotroph organisms, particularly on cropland, where a major decline was observed for UhB, both below and above ground. The drop of this indicator on cropland reflects the declining state of biodiversity associated with changes undergone by Spanish agriculture [51,70].

### 3.2. Restoring the Internal Loops in Spanish Agrolandscapes

Each landscape structure configures a metabolic arrangement which in turn conditions the ecological processes (energy and material flows, natural population regulation, etc.) in the agroecosystem. The restoration of internal loops in Spanish agriculture would require a major transformation of the landscape matrix and a redesign of the crop and livestock systems. However, the necessary redesign of the territory requires knowing the territory needed to sustain the Spanish population in a sustainable manner beforehand. In this regard, a study was performed by a research team at the Universidad Pablo Olavide in Spain quantifying the land cost of the transformation of Spanish agriculture and livestock farming to organic production, assuming that the Spanish territory would provide the nitrogen flows and functional biodiversity necessary to allow the functioning of the agroecosystems [56]. The land cost was evaluated in two different scenarios. In scenario 1, all conventional Spanish agriculture is transformed into organic farming, adopting current organic farming

practices and yields. Scenario 2 is based on scenario 1 but seeks to model the land cost of intensive organic farming based on low-entropy internal loops. This requires practices such as the sowing of green legume manures and the use of agroindustrial waste as fertilizer. In addition, it supposes the development of integrated crop/livestock systems, where animals are fed hay, grain, and by-products from food production, and in turn, provide manure for the most demanding crops.

Figure 4 shows the increase in total crop area and changes in production orientation that would be required under scenario 1. The crop area would have to grow to 20 Mha (2.7 million more than the area available in 2008), fallow land would disappear, and the extra crop area devoted to ecological infrastructure would approach a million hectares, improving the landscape and connectivity of crops with areas of natural vegetation. The largest growth would be seen in winter herbaceous crops (including barley), which would multiply the current crop area by 1.5.

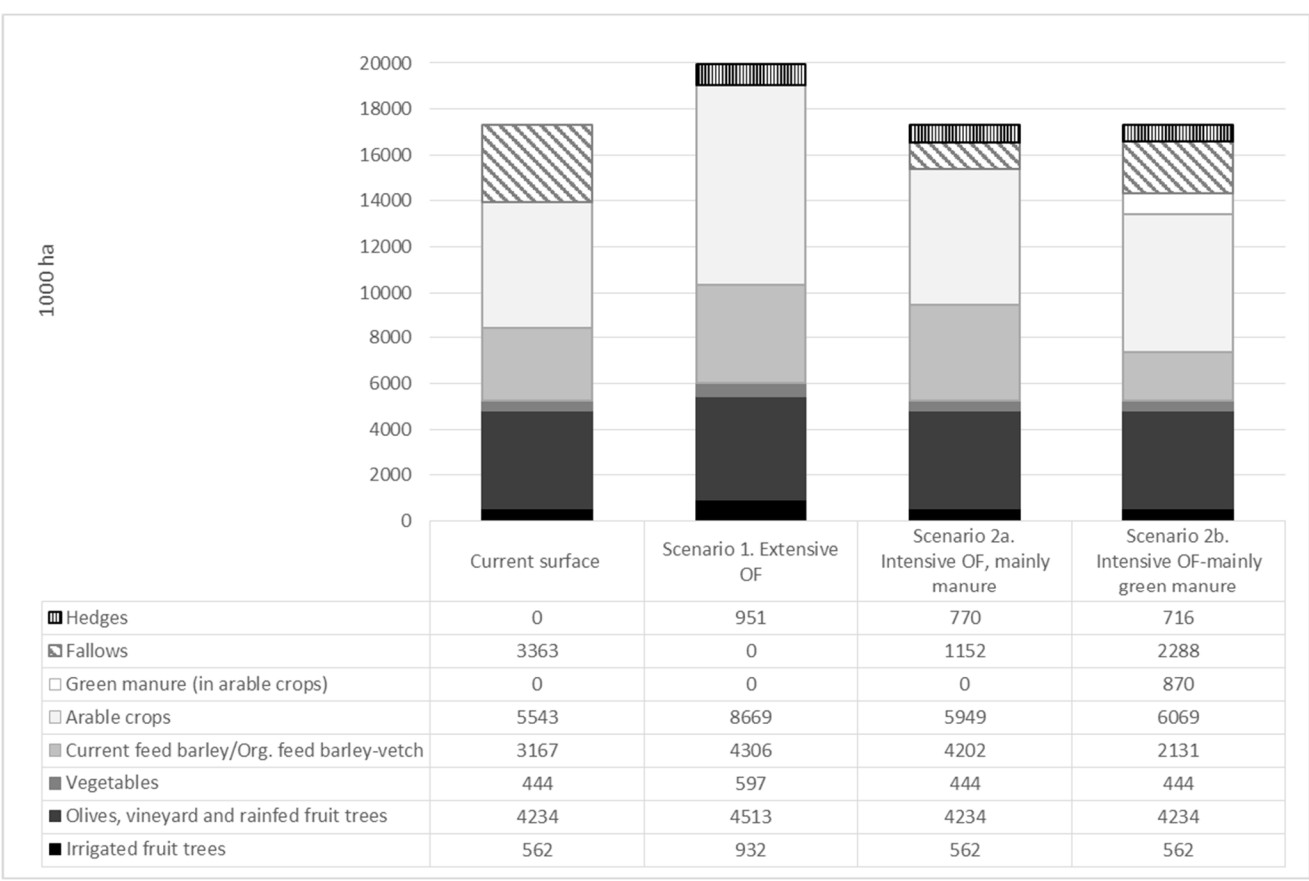

| | Current surface | Scenario 1. Extensive OF | Scenario 2a. Intensive OF, mainly manure | Scenario 2b. Intensive OF-mainly green manure |
|---|---|---|---|---|
| ▥ Hedges | 0 | 951 | 770 | 716 |
| ◩ Fallows | 3363 | 0 | 1152 | 2288 |
| ▫ Green manure (in arable crops) | 0 | 0 | 0 | 870 |
| ▫ Arable crops | 5543 | 8669 | 5949 | 6069 |
| ▨ Current feed barley/Org. feed barley-vetch | 3167 | 4306 | 4202 | 2131 |
| ▪ Vegetables | 444 | 597 | 444 | 444 |
| ▪ Olives, vineyard and rainfed fruit trees | 4234 | 4513 | 4234 | 4234 |
| ▪ Irrigated fruit trees | 562 | 932 | 562 | 562 |

**Figure 4.** Cropland distribution at present and in different organic scenarios by crops (excluding 80,924 ha of non-food crops). Source: [70].

In scenario 2, the conversion from conventional to organic farming alone (17.4 Mha) would mean a fall of 13% in Spanish agricultural production. However, it would not be necessary to increase the crop area due to changes in the orientation of production and more intensive farm management, reducing the land cost of conversion to organic farming. With increased manure input for nitrogen fertilization (see scenario 2a in Figure 2), fallow would be reduced by 2.2 Mha; the area would be used for feed barley and hay vetch (1 Mha), arable crops (green manure seed production, 0.4 Mha), and hedges around fields (0.8 Mha). With a higher contribution of green manure (see scenario 2b in Figure 4) instead of animal manure, fallow would only be reduced by 1.1 Mha. A similar reduction (1 Mha) would be seen in the crop area of feed barley and hay vetch, which, in this scenario, would not be so necessary to provide flows of N for fertilization in organic farming. In contrast, there

would be an increase of 0.9 Mha in the crop area devoted to green manure and 0.5 Mha devoted to arable crops (green manure seed production). Such management would allow the area occupied by hedgerows and other seminatural habitats to rise to 0.7 Mha, thus enhancing the provision of biological control and pollination services to crop production.

## 4. Conclusions

The conversion of agroecosystems to agroecological management is linked to the positioning of the agroecosystem and its connectivity relationships with the different types of surrounding natural and semi-natural habitats. Understanding the spatial and functional organization of the landscape matrix in interaction with neighboring agroecosystems is essential to promote patterns and mechanisms that foster biodiversity and the provision of multiple ecosystem services. For example, it is well documented that the effects of plant diversification on insect pest populations are mediated at the landscape level because diversified surrounding natural and semi-natural vegetation benefits natural enemies and thus the biological control of pests. In this regard, the ABIHS methodology is a useful tool to assess whether the configuration and botanical composition of the agrolandscape are conducive to enhancing biocontrol, and if not, what agroecological plant compositional and configurational designs may be needed to improve habitat quality for beneficial insects, as shown in the comparison of two California vineyards.

MAS complements ABIHS by providing metrics on the composition, configuration, and heterogeneity of the landscapes that encircle agroecosystems, yielding critical insights to be considered when designing agroecosystems in agroecological transition. In the Chilean case study, MAS allowed us to observe the relationship between landscape structure, the presence of native vegetation patches, and the response this generates to natural enemies. The data generated reinforced the need to preserve and even increase areas of natural and semi-natural vegetation within the agroecosystem and its surrounding perimeter.

Spanish agriculture's industrialization led to the loss of functional synergies typical of integrated agrosilvopastoral systems, limiting the replenishment of soil fertility with animal manure, thus increasing the use of chemical fertilizers while diminishing the energy efficiency of agroecosystems. The advance of monocultures reduced forest lands compromising biodiversity levels and farm total productivity. SAM methodological analysis suggests that restoring the agrolandscape structure via organic farming featuring animal integration and the use of green manures would allow substantial restoration of areas occupied by hedgerows and other seminatural habitats, thus creating positive conditions for the provision of biological control and pollination services to crop production.

The three methodologies can be applied simultaneously to assess agrolandscape-level interactions in situations of regional agroecological transition. The challenge lies in the capability of research teams to implement the approaches at the field level and integrate the various indicators to assess relationships between the habitat diversity and ecosystem services linked to beneficial insects and the material and energy flows between agroecosystems and surrounding environments at a regional level. Results from the analysis can inform landscape planning for agroecological transition to promote robustness and resilience towards climate change while restoring biodiversity for agricultural productivity and establishing ecological networks in the landscape.

**Author Contributions:** Conceptualization and methodology, M.A.A., M.G.d.M. and A.S.R.; writing—original draft preparation, M.A.A., C.I.N., M.G.d.M. and A.S.R.; writing—review and editing M.A.A., C.I.N. and A.S.R. All authors have read and agreed to the published version of the manuscript.

**Funding:** This research was supported by funds from each researchers' program.

**Data Availability Statement:** No new data were created or analyzed in this study. Data sharing is not applicable to this article.

**Acknowledgments:** We gratefully acknowledge the collaboration of participating farmers in Chile, California, and Spain. The work of Angel Salazar-Rojas in this study has been supported by the Anillos Projects ANID/230028.

**Conflicts of Interest:** The authors declare no conflicts of interest.

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
