# Peer review of "Landscape Agroecology: Methodologies and Applications for the Design of Sustainable Agroecosystems"

_land, doi:10.3390/land13111746_

Round 1

Reviewer 1 Report

Comments and Suggestions for Authors

This article is extremely interesting and will be an important contribution to research demonstrating the benefits of biodiversity conservation for agriculture. The methodologies tested will surely be of great value to practitioners and researchers. 

 General comments:

-          It may be worth acknowledging the problems with concepts such as “ecosystem services” in the context of agroecological movements and Indigenous worldviews.

-          The introduction introduces two of the methodologies only.

-          Sometimes SAM is used and sometimes AM. It would be helpful if this could this be clarified or just one acronym used.

Specific comments:

-          In the first paragraph, it would be helpful for the reader if examples of ecosystem services for crop productivity were identified briefly. Even though they are discussed later (page 2), it seems like a missing element in this first paragraph.

-          Line 71: consider adding “external” inputs

-          Line 139, typo: “schemes”

-          Line 268, typo: “parasites”?

-          Line 320: Sentence needs clarification. Is the new methodology mentioned still SAM?

-          Line 357: “tripled”

-          Line 406: The study was carried out by the authors, correct? It may be useful to separate the sentence into two for greater clarity as to what is being cited.

Author Response

  • We made clear in the introduction that we are describing three methodologies
  • Throughout the manuscript we made sure that SAM refers to Social Agrarian metabolism
  • We corrected all the queries and typos in lines 71, 139, 268, 320 and 357
  • We could not fit in the text the suggestion provided by reviewer 1 “It may be worth acknowledging the problems with concepts such as “ecosystem services” in the context of agroecological movements and Indigenous worldviews” as we don’t see the relationship of such statement with the concepts we address in the manuscript.    

Reviewer 2 Report

Comments and Suggestions for Authors

This paper is a review describing 3 methods for assessing the sustainability of agroecosystems, at the scale of the landscape. The purpose is to combine them in a comprehensive approach for designing farming systems, or for helping territories with their agroecological transition.

The rationale of the paper is systems thinking, based on the concept of ecosystem services, with farms in relationship to their environment. It relates farmed surfaces (crops, feed and pastures) with natural or seminatural elements of the landscape. The overall organization at the landscape level is described as farms in an environmental matrix, with patches that are more or less connected and diverse in size, landcover and landuse.

The layout of the paper is to present the 3 methods in 3 case studies, first the “Assessment of Beneficial Insect Habitat Suitability - ABIHS” on two vineyards in Northern California, second the “Main Agroecological Structure - MAS” on Chilean family farms, and third the “Agrarian Metabolism - AM” on Spanish agricultural landscapes.

This paper is both timely and valuable, providing researchers and stakeholders with an overview of complementary and practical assessment methods at the landscape scale. Landscape planning for agroecological transition is an urgent issue all over the world. It is the right scale to promote robustness and resilience towards the erratic meteorological conditions that are more and more frequent with climate change. It is also the saving strategy for restoring biodiversity in these human-managed ecosystems. Here, the focus is on promoting biodiversity for agricultural productivity (ecosystem services) but more generally, the good functioning of ecological networks in the landscape is vital for all socio-ecosystems.

The concepts underlying this review are well documented, with sound scientific evidence: first on the relationship between the diversity of habitats and ecosystem services linked to beneficial insects ; second on landscape metrics accessible to farmers and landscape planners, that are linked to multiple ecosystem services, especially facing climate change ; third on material flows between parts of the agroecosystems at a regional level, which are one-directional in intensive farming and detrimental to non-domestic species, whereas agroecological farming is more integrated and balanced. In the third approach, social and economical aspects are taken into account.

However, the structure of the paper is quite confusing and should be revised, to qualify as a review. Indeed, the abstract does not clearly announce that this review compares 3 complementary methods of assessment, illustrating each with a different case study. This should be stated clearly also at the end of the introduction (before subchapter 1.1.).

At a first glance, the titles of the chapters and sub-chapters reflect the layout in three parts, but in detail, their wording and contents are very different and inconsistent between levels.

1.      “Introduction” : the following paragraphs are a general overview of landscape and biodiversity, then in the subchapter 1.1., there is more detailed information about the link between natural habitats and biodiversity in cultivated areas, and the subchapter 1.2. presents specifically the ABIHS method and illustrates it with a first case study in Northern California

2.      “The Main Ecological Structure” : the following paragraphs present the MAS method, the subchapter 2.1. applies it to a second case-study (Chilean farms) and the subchapter 2.2. is a review of the responses of natural enemies to some of the indicators that are assessed with the MAS method.

3.      “A metabolic approach to agricultural landscapes” : the following paragraphs present the principles of Social Agricultural Metabolism, then the subchapter 3.1 is called “The landscape as socio-metabolic footprint” and shows how the indicators of this method are built, on the example of the history of Spanish agricultural specialization and intensification. The subchapter 3.2. presents 2 scenarios of restoration of sustainable systems, by strengthening internal “metabolic” loops.

The first paragraphs of the introduction should be taken out of chapter 1 and stand alone as a general introduction. For consistency, the 3 chapters could be named after the intention of each method of assessment, e.g. : “2. Assessing the landscape features in and around the farm.” The subchapters would then be 2.1. What do we know? make a bibliographic review of the knowledge about the link between landscape metrics and biodiversity 2.2 How do we assess with the MAS tool? Give details about this assessment method and apply it to the case-study. Or the other way round, but consistently for the 3 chapters.

A major concern: throughout these 3 parts, it is unclear if the case studies are cited from literature, or if they are original to this review. If they are original, there is little information about the source of data. Figure 1 and Table 1 don’t mention any bibliographic reference, and although figure 4 does cite one reference [70] Guzman et al., the graph that it shows does not appear in the source, nor seems to be derived from their data.

As a conclusion, the scientific content of this review is sound and very useful, but the inconsistent structure makes it very difficult to read. Illustrating each assessment method with a case study is an interesting approach, but it is unclear if the case study results come from published papers, or have been generated for illustrating this review. It would also have been interesting to state in the conclusion if it is possible to use these 3 approaches simultaneously in a situation of regional agroecological transition.

Author Response

  • We mention in the abstract that we present three methodologies that can be applied simultaneously to assess agrolandscape level interactions in situations of regional agroecological    transition.
  • We provided the source of Figures1, 2, 3 and 4 and when there is no source for the data, we specified (Altieri and Nicholls, unpublished data or Salazar Rojas, unpublished data).
  • As suggested by reviewer 2, we renamed subsections 2 and 3 after the intention of each method of assessment.
  • Reviewer 2 suggested “The first paragraphs of the introduction should be taken out of chapter 1 and stand alone as a general introduction” but we could not understand the purpose of such suggestion. We think the introduction is clear by presenting the three methodologies as tools to analyze and assess the ecological relationships between agricultural systems and the surrounding matrix. Such information is key for the design of sustainable agroecosystems that are resilient and dependent on ecological processes rather than external inputs in the context of an agroecological transition at the territorial level.
  • We feel that paragraphs contained between lines 131-140 need to remain within section 1.2 because it justifies the need for the ABIHS methodology to assess if the crop/habitat composition and configuration of a given agrolandscape is conducive to enhance biocontrol of pests.
  • We cited a key reference (see reference 29) for interested readers to delve in more detail on the MAS methodology and how it is used to assess the link between landscape metrics and biodiversity.